# Reanalysis of Published Histological Data Can Help to Characterize Neuronal Death After Spinal Cord Injury

**DOI:** 10.3390/ijms26083749

**Published:** 2025-04-16

**Authors:** Pablo Ruiz-Amezcua, Nadia Ibáñez-Barranco, David Reigada, Irene Novillo, Altea Soto, María Asunción Barreda-Manso, Teresa Muñoz-Galdeano, Rodrigo M. Maza, Francisco J. Esteban, Manuel Nieto-Díaz

**Affiliations:** 1Molecular Neuroprotection Group, National Hospital for Paraplegics (SESCAM), Instituto de Investigación Sanitaria de Castilla-La Mancha, 45071 Toledo, Spain; prazmezcua@sescam.jccm.es (P.R.-A.); ibnadia99@gmail.com (N.I.-B.); dreigada@sescam.jccm.es (D.R.); irenenoal@outlook.com (I.N.); alteasotoneira@gmail.com (A.S.); mbarreda@sescam.jccm.es (M.A.B.-M.); tmunozd@sescam.jccm.es (T.M.-G.);; 2Department of Experimental Biology, University of Jaén, 23071 Jaén, Spain; 3Global Services Learning Associate Human Pharma Regions in Boehringer Ingelheim, 08174 Barcelona, Spain; 4Functional Exploration and Neuromodulation of Nervous System Investigation Group, National Hospital for Paraplegics (SESCAM), Fundación del Hospital Nacional de Parapléjicos para la Investigación y la Integración, Instituto de Investigación Sanitaria de Castilla-La Mancha, 45071 Toledo, Spain

**Keywords:** spinal cord injury, animal models, neuronal death, neuroprotection, image analysis, image registration, open science

## Abstract

Neuronal death is a central event in spinal cord injury (SCI) pathophysiology. Despite its importance, we have a fragmentary vision of the process. In our opinion, the research community has accumulated enough information to provide a more detailed, integrated vision of neuronal death after SCI. This work embeds this vision by creating an open repository to store and share data and results from their analysis. We have employed this repository to upload raw images of spinal cord sections from a mouse model of contusive SCI and used this information to compare manual-, threshold-, and neural network-based neuron identifications and to explore neuronal death at the injury penumbra 21 days after injury and the effects of the anti-apoptotic drug ucf-101. Results indicate that, whereas the three identification methods assayed yield coherent estimates of the total number of neurons per section, neural network (NN) outperforms the other two methods. Combining NN identification and image registration has allowed us to characterize neuron distribution among Rexed laminae in the mice T11, revealing spatial patterns in the neuronal death that follows injury and in their survival following ucf-101 treatment.

## 1. Introduction

According to the WHO [1], the term SCI refers to the damage to the spinal cord resulting from either external physical impact (traumatic SCI) or from disease and degeneration (non-traumatic SCI). However, there is no reliable estimate of global prevalence. Up to 7 million SCI patients in the world [2] live with permanent disabilities of enormous personal, medical, social, and economic costs ($1.5 to 6.2 million/patient in the USA, according to [3]). SCI leads to extensive alterations in the central nervous system (CNS), impairing motor, sensory, and autonomic functions. Research over the past 40 years has yielded multiple therapies, several of which have been tested in clinical trials [4]. However, to date, early surgery interventions, intensive care management, and rehabilitation remain the cornerstones of SCI treatment [5,6].

Trauma to the spinal cord triggers a complex pathophysiological cascade that extends the initial damage to distant regions of the CNS during the following weeks and months [5]. Cellular death, particularly neuronal death, is a central event in SCI pathophysiology and a major determinant of the resulting functional deficits. In traumatic SCI, neuronal death is biphasic. Initially, it was necrotic, driven by the physical breakage of axons and somas. This is followed by a cascade of pathophysiological events that alter the spinal cord environment, triggering secondary waves of neuronal death that extend throughout the spinal cord and the brain [7]. Contrary to the necrotic death that characterizes the primary injury, secondary damage is characterized by programmed processes [8,9,10,11], which might be regulated with appropriate neuroprotective treatments [9].

Histological analyses back in the 1990s and the early 2000s provided spatial and temporal characterizations of neuronal death in animal models of SCI, describing the onset and spread of necrotic and apoptotic processes (see, for example, Beattie et al. [12]). Since then, advances in knowledge, technologies, and techniques have allowed the identification of novel forms of Regulated Cell Death (RCD) acting on the injured spinal cord, as well as characterizing the effects of multiple neuroprotective treatments (for review, see Shi et al. [13]). In our capability of managing and analysing increasingly large quantities of data, recent developments have opened the opportunity to study the different processes affecting spine cells with single-cell resolution [14] or to reconstruct the cell architecture of the spinal cord [15]. Integrating these tools with the histological data accumulated over decades of research would provide a precise understanding of the temporal and spatial progression of neuronal death after SCI. This could help identify its causes, mechanisms, and associated processes, as well as improve our understanding of the effects of neuroprotective treatments. However, such integration is still lacking, and to our knowledge, no comprehensive characterizations of neuronal death have been published since studies from 20–30 years ago.

Here, we hypothesize that the histological information accumulated over the years can be reanalysed using novel techniques to help fill this gap. In this respect, this work tries to contribute by:Creating an open data repository to store and share images, as well as the results from their analysis;Depositing and documenting 168 images of full transversal sections of mice spinal cords from the study by Reigada et al. [16] on the neuroprotective effects of ucf-101 in the damaged spinal cord;Comparing the precision and reproducibility of manual-, threshold- and neural network-based neuron identification methods;Exploring how neuronal death distributes within the spinal cord sections in Reigada’s mouse model of SCI and which neurons become protected by the treatment with the anti-apoptotic drug ucf-101.

## 2. Results

### 2.1. Open Access Repository

We have employed Open Science Framework (OSF, https://osf.io; accessed on 7 April 2025 [17]) to store all images, data, and results (see table below) from this article. All contents are freely available under the Creative Commons (CCO 1.0 Universal) license. The project, named neuroCLUEDO|Spinal Cord Injury (https://osf.io/n32z9/; accessed on 7 April 2025), is aimed to determine which, where, and how neurons die after SCI. The information concerning image analysis is distributed in three major components:A documented image repository of spinal cord sections stained with neuronal markers.Test benches for comparing methods to analyse naïve and injured spinal cords.Results from the analysis of the images in the neuroCLUEDO repository.

All raw and processed images, together with any additional information described here and the results from the comparison of neuronal detection methods and the reanalysis of the effects of ucf-101 on neuronal survival after SCI, have been distributed among the three components of neuroCLUEDO as detailed in Table 1.

### 2.2. Comparison of Methods for Identifying Neurons

#### 2.2.1. Neuronal Identifications

We have compared the neurons identified through manual, threshold-based, and neural network-based methods in 20 spinal cord sections originally studied by Reigada et al. [16]. The identifications obtained through the three methods can be accessed at the neuroCLUEDO repository (https://osf.io/7agyq/, accessed on 7 April 2025).

As shown in Figure 1, each method yields different outputs. Manual identifications result in the X and Y coordinates of each neuron, thresholding results in a binary segmentation of the image, whereas the neural network assigns a probability of being part of a neuronal nucleus to each pixel in the image. Despite these differences, the three methods yield broadly comparable identifications, all within the grey matter—an established condition in manual and threshold-based identifications but not in neural network-based ones. In addition, the neural network and the manual methods were able to exclude most artifacts, whereas the threshold method often failed to discriminate them from neurons.

To explore the coherence of the identifications provided by the three methods, we overlapped the objects identified as neurons by each method in each image. In the case of manual identification, there were three or five analyses (from two or three observers, respectively; see Appendix A) per image, which were overlapped—each object was characterized by the number of analyses in which it was identified as a neuron. As shown in Figure 2, most neurons were identified in a single manual analysis and can therefore be considered highly doubtful (see also Appendix A). These doubtful neurons are particularly abundant in sections of untreated injured cords. The objects identified as neurons by all analysts, which therefore can be considered undoubtedly as neurons, were also abundant, particularly in sections from control individuals.

To compare with the threshold-based and the neural network identifications, we overlapped the objects identified as neurons by each method in five images (all with five manual analyses). As Figure 2 illustrates (see details in Appendix A), threshold and neural network methods detected most of the objects identified as neurons in three, four, or five manual analyses. Conversely, those objects with less agreement among manual identifications (1 or 2 out of 5) showed markedly less agreement with the threshold- and neural network-based identifications, confirming that assignment to neurons is at least doubtful. These results suggest that both the macro and the neural network are able to identify as neurons those objects with the highest probabilities of belonging to this cell type and only show greater discrepancies with the human analysts in doubtful cases. On the other hand, both the macro and the neural networks are identified as neuron objects that are not considered as such by any of the human analysts (value 0 in Figure 2; see also Appendix A). The number of these objects is higher in the threshold results (82% additional cells) than in the case of the neural network (19%). Assuming that these objects correspond to artifacts and not to neurons, it can be deduced that the neural network is capable of quite effectively discriminating cells with high signal due to autofluorescence or artifacts that do not correspond to neurons, while the threshold-based method has difficulties in this aspect. Additionally, both methods have problems in separating some cells that are very close to each other, and the threshold method, in particular, sometimes separates objects arbitrarily.

#### 2.2.2. Number of Neurons

The number of neurons in a defined region is one of the parameters derived from the identifications that are most commonly used in scientific studies of SCI and other pathologies of the nervous system. To evaluate the neuron identification methods, we compared the repeatability, reproducibility, and accuracy of neuron counts recorded in 20 sections (see online file at https://osf.io/6mnb4; accessed on 7 April 2025). We did not evaluate the Neural Network method’s repeatability and reproducibility because we assume its identifications are 100% repeatable and reproducible.

Repeatability: Repeatability was studied by comparing two neuronal counts from the same image recorded blindly by an observer/operator with at least one week span between measurements. For evaluation, the difference between both values was represented against their average, following Bland and Altman [19]. The resulting graphs (Figure 3) show important changes in the number of neurons estimated in the repeated measurements from both the manual (95% confidence interval (C.I.): +111.38; −138.78; *n* = 30) and threshold-based (95% C.I.: 140.36; −154.6; *n* = 100) methods.

Various factors can influence repeatability. We have considered four of them to possibly influence the repeatability in analyses: mean number of neurons, time to conduct the analysis, experience of the analysts, and condition of the spinal section. To assess their explanatory potential, we adjusted linear models relating the difference between the repeated counts (inverse of repeatability) in a section with the four variables. In the analysis of repeatability of manual data, the comparison of the fitting of the resulting models using analysis of variance (ANOVA) showed that the best fit is achieved by the model that only employs the time as a predictor (Adjusted R^2^ = 0.504, *p* < 0.001, *n* = 30), suggesting that the more effort required to identify the neurons in an image the lower the repeatability. For threshold-based data, the best fit was achieved by a model including the mean number of neurons and the condition (Adjusted R^2^ = 0.338, *p* < 0.001, *n* = 100), revealing that higher repeatability is achieved in sections with lower numbers of neurons and in those from uninjured spinal cords (see online file at https://osf.io/6mnb4; accessed on 7 April 2025).

Reproducibility: Reproducibility of manual and threshold-based identifications was studied by comparing the neuronal counts estimated for the sections by the different analysts. As can be seen in the Bland–Altman graph (Figure 4), reproducibility was similar to repeatability, only with a slight increase in the 95% confidence intervals (+143.87, −162.25 for manual analyses, *n* = 40; 125.7, −112.44 for threshold analyses, *n* = 200).

As in the repeatability tests, we further analysed the effect of the number of neurons and the type of section (treatment and distance from injury epicenter) on reproducibility through linear models and compared them using ANOVA. In both cases (manual and threshold identifications), the best fits were achieved by models predicting the effects of the mean number of neurons and the treatment on reproducibility (manual identifications: adjusted R^2^ = 0.559, *p* < 0.01, *n* = 40; threshold identifications: adjusted R^2^ = 0.376, *p* < 0.01, *n* = 200) (see online file at https://osf.io/6mnb4; accessed on 7 April 2025).

Accuracy: The accuracy of the manual, threshold-based, and neural network-based neuronal counts was estimated by comparing the estimated values of neurons in each of the images of the comparison set against the Reference Number of Neurons (RNN) derived from the weighted consensus of the manual estimations for each image (Figure 5 and Appendix A).

As shown in Figure 5, manual counts show a high correlation (Pearson’s correlation coefficient (r2) = 0.926, *p* < 0.001, *n* = 50), with all individual analysts showing broadly comparable correlations. The threshold- and neural network-based methods also result in values highly correlated with the RNN (see Figure 5). The threshold-based method presents a slightly lower correlation (correlation coefficient = 0.816, *p* < 0.001, *n* = 20) compared to the neural network identifications (correlation coefficient = 0.864, *p* < 0.001, *n* = 20). Therefore, despite this difference, it can be considered that the values obtained in the article by Reigada et al. [16] using this threshold method are an acceptable estimate.

Processing time: The time of analysis was incorporated in this study as a measure of the effort made to identify the neurons in an image. For manual identifications, the average time required to analyse an image has been 17 min and 40 s with a standard deviation of 11 min plus 58 s and a range from 39 s to 67 min (see the online file at https://osf.io/6mnb4; accessed on 7 April 2025). On the other hand, the time spent in the threshold-based identifications was 1 min plus 55 s per image (Standard Deviation = 36 s) while, using the neural network developed in this work, the identifications of the 20 images required 3 min in total. In both cases, the time to develop the neural network and the macro for threshold-based identifications should be added to the processing time. In the case of the neural network, it was necessary to invest 15 h of work for training and preparation, whereas the design and preparation of the threshold-based macro required more than 30 h of work.

### 2.3. Effects of Injury and ucf-101 Treatment on Neuronal Death

#### 2.3.1. General Description of the Obtained Data

In this second part of the study, we evaluated the effects of SCI and treatment with ucf-101 on the extent and spatial distribution of neuronal death in the spinal cord. We focused our analysis on the sections 0.6 to 1.2 mm caudal to the injury epicenter, where Reigada et al. [16] observed significant neuroprotective effects of ucf-101. The images from 41 sections (available at https://osf.io/fhxqs/; accessed on 7 April 2025) were processed to identify and quantify the number of neurons in each Rexed laminae. Alterations in the sections, staining failures, and other problems precluded the analysis in four of these sections. Therefore, 37 sections were included in these analyses (see Appendix A). The final data matrix comprising the number of neurons in each Rexed laminae per section is available at OSF (https://osf.io/bhg6f; accessed on 7 April 2025).

In undamaged individuals, a median of 282.5 neurons is recorded in four spinal cord sections, although 503 neurons were recorded in one section. Despite this variability, patterns of neuron distribution among Rexed laminae and nuclei are clearly appreciable. As shown in Figure 6 and Appendix A, the highest numbers of neurons are present in the dorsal laminae, particularly in laminae 3 (L3) and 4 (L4), and to a lesser extent in the lower part of lamina 7 (L7b). On the contrary, minimal counts are observed in lamina 9 as well as in the intercalated (ICI), intermediolateral (IML), or dorsal Clarke’s (D) nuclei. To evaluate the congruence of image registration, we compared the number of neurons in each laminae and nuclei obtained here with the numbers derived from the registered image of the T11 segment of a P56-day-old mouse from the reference Atlas for the mouse spinal cord available at the Mouse Spinal Cord Atlas by Watson et al. [18]. As shown in Appendix A, the percentages of neurons per region (lamina or nucleus) obtained in our analyses show good agreement with those derived from the Atlas, with only the values from L1, L5l, and intermediomedial (IMM) nucleus showing discrepancies above twofold. Conversely, the absolute number of neurons per lamina is highly divergent, as could be expected considering the differences in tissue processing and thickness.

#### 2.3.2. Neuronal Loss After Spinal Cord Injury

To characterize the neuronal losses 21 days after injury in the present mouse model, we compared the median neuronal counts in sections from uninjured individuals with the counts of sections 0.6 to 1.2 mm caudal to the injury epicenter from six individuals that underwent a moderate contusion (50 Kdynes) at spinal segment T11. Examination of the data first reveals a strong variability in the number of preserved neurons at each distance. As an illustration, sections 0.6 mm away from the injury epicenter comprise one without neurons, a second with 6 neurons, and a third with 122 neurons. Variability—likely representing differences in injury severity or failures in precisely identifying the injury epicenter—difficult drawing conclusions on the neuronal death patterns. Nevertheless, the data clearly reveal that compared to the median of 283 neurons recorded in the sections of the uninjured individuals, the sections from injured individuals show a reduction in the number of neurons in all laminae at the four distances under study. Neuronal loss is massive in the sections 0.6 mm away from the epicenter (median total number of neurons = 6.5, ~2.3% survival) and recovers gradually with a pace of nearly 20% neurons every 200 microns to reach a maximum of 40% survival at 1.0 mm without showing further recovery at 1.2 mm (see Appendix A). At the individual level, the profile generally shows a sustained increase in distance to the epicenter. An increase in surviving neurons is also observed in all laminae and nuclei with some degree of heterogeneity. Dorsal laminae, particularly L2 to L4, show high neuronal survival even in sections close to the epicenter, reaching above the 90% of control values at 1.0 and 1.2 mm away from the epicenter in some individuals. High neuronal survival is also observed in L9 and the IML, although the lower number of neurons recorded in these regions may bias the results. Conversely, survival is particularly poor in laminae L5l, L5m, L8, and L10, as well as in D, ICI, and Lumbar Dorsal Commissural (LDCom) nuclei, all in the intermediate gray substance. Leaving apart the individual 3U, which presents exceptionally high values at all analysed distances, neuronal survival at 1.2 mm away from the epicenter is below 60% in all sections in these regions. Kruskal Wallis analyses (shown in Appendix A) confirm the highly significant losses of neurons in multiple laminae and nuclei from sections sampled 0.6 and 0.8 mm caudal to injury, whereas at 1 and 1.2, losses are non-significant. Despite general trends, variability among individuals within the same condition is high (see violin plots in Figure 6), which may preclude statistical significance.

#### 2.3.3. Neurons Protected by ucf-101

Analyses from Reigada et al. [16] revealed that treatment with ucf-101 results in a localized neuroprotection centered in the penumbrae segments 0.6 to 1.2 mm caudal to the injury epicenter. Focusing on these sections, our results confirm ucf-101 neuroprotection in the whole sections in this region; however, contrary to Reigada et al. [16], protection appears restricted to the sections closer to the injury epicenter (0.6 and 0.8 mm caudal to the injury epicenter; see Appendix A) while the numbers of surviving neurons become even at 1 mm caudal to the epicenter. Data from 1.2 mm are too sparse in the ucf-101 group to draw any significant observations. A deeper exploration of the distribution of ucf-101 neuroprotection among the neurons in the different Rexed laminae and spinal cord nuclei (Appendix A and Figure 6) reveals that ucf-101 neuroprotection is heterogeneous and appears to concentrate in specific laminae and nucleus of the intermediate region (laminae L4, L5l, L7, and nuclei D, and ICI), interestingly leaving out lamina L5m. These ucf-101 protective effects appear to be consistent across the examined distances to the injury epicenter, whereas other effects, such as the protection in laminae L1, L2, and L9 or the nucleus IML, are only observed at specific distances (Figure 6).

## 3. Discussion

Neuronal death following SCI is a complex and multifaceted process. Research suggests that different types of spinal neurons may indeed exhibit varying susceptibilities to secondary degeneration. While there is evidence to support these differences [7], it is important to note that the exact mechanisms and extent of susceptibility can be influenced by various factors, including the location and severity of the injury, as well as the specific animal model used for research.

Recent technological breakthroughs—including image analysis and -omics technologies with single-cell resolution, tools for processing big data, and availability of repositories capable of storing and sharing gigabytes of information—promise to revolutionize Biomedicine [20]. The present study stems from this background to develop a framework to integrate histological and -omics information and tools for the analysis of neuronal death after SCI. Here, we focus on histological images of SCI already available in the laboratories, which may be a key resource for the analysis of SCI. To this aim, we have created an open-access repository that stores 168 images of transversal sections of mice spinal cords from the study by Reigada et al. [16] on the neuroprotective effects of ucf-101 in the damaged spinal cord. The images have been employed for two goals: (i) to demonstrate the outperformance of neural networks for neuron identification in histological preparations of the damaged spinal cord; (ii) to explore how neuronal death is spatially distributed in a mouse model of contusive SCI and which neurons become protected after treatment with ucf-101 anti-apoptotic drug.

### 3.1. Repository on Spinal Cord Injury

The primary contribution of this study is the establishment of an open-access repository, known as neuroCLUEDO|Spinal Cord Injury, hosted on the OSF. This repository contains a comprehensive collection of histological images obtained from animal models of SCI, documented in accordance with the MIASCI standard [21]. To the best of our knowledge, neuroCLUEDO represents the first open-access repository of histological images from animal models of SCI. It encompasses full transverse sections from 13 individuals with and without SCI, and we are committed to continuously expanding this repository with additional individuals.

Moreover, neuroCLUEDO serves as a comprehensive archival resource, housing not only the raw images but also all processed images, data, and results derived from image analyses. Additionally, it provides access to the protocols, scripts, and tools employed, all of which are readily available for exploration, utilization, or comparison by the research community. While the current contents of the neuroCLUEDO repository primarily reflect the outcomes of this study, our intention is to broaden its scope by incorporating images from other investigations and facilitating comparisons of various methodologies. This expansion encompasses aspects ranging from staining techniques to image acquisition and registration processes and aims to contribute to our understanding of neuronal death following SCI, the effects of treatments, and the vulnerabilities of distinct neuronal populations.

Furthermore, we aspire to integrate ‘-omic’ data, tools, and results, with a particular emphasis on single-cell or single-nucleus RNA sequencing data and the means to combine and analyse both data sources. Our overarching objective is to facilitate collaboration, knowledge sharing, and advancements in the field of SCI research through this comprehensive and continually expanding repository.

### 3.2. Test Bench for Neuronal Identification Methods

In its maiden application within the neuroCLUEDO repository, we have embarked on a comparative analysis of neuronal identification methods, namely manual, threshold-based, and neural network-based approaches. The purpose is to evaluate their potential and limitations. Within the repository, we have established a test bench that is accessible to all for conducting further comparisons. Our current comparisons underscore the superiority of the neural network method over the manual and threshold-based methods.

Traditional manual methods have been a prevalent choice for neuronal identifications, probably due to their simplicity and minimal technological requirements. However, our results suggest that manual methods are the most time-consuming and exhibit limited repeatability and reproducibility. The latter two drawbacks are of particular concern, given the substantial disparities observed, sometimes exceeding 100 neurons in repeated measurements of sections with fewer than 200 neurons. This leads to the observation that the most commonly identified neurons occur in only one out of every five analyses. Even with an increase in expertise, our analyses reveal that experienced researchers struggle to consistently deliver repeatable identifications. Indeed, as a result of our study, our lab opted to duplicate the manual analyses for each image to enhance identification certainty.

The second method, reliant on thresholding to establish staining levels of neuronal and nuclear markers for neuron identification, represents an alternative approach [16,22]. The thresholding procedure here employed is semi-automated, requiring manual selection of the thresholding algorithm due to substantial variations in background and staining intensity across different preparations and even within sections of the same preparation. Consequently, repeatability and reproducibility issues arise due to differences in the chosen thresholding algorithm. While this method offers time-saving benefits over manual approaches, its efficiency is constrained. Notably, the selected thresholds often lack specificity, resulting in the inclusion of objects that human observers do not identify as neurons. To address this issue, we integrated a manual selection of gray matter within the macro to exclude objects identified outside of it [16]. Improvements in thresholding techniques may be achievable through advancements in image preparation, background correction, and the selection of thresholding algorithms, among other factors. However, the extent to which these efforts are justifiable remains a topic of debate, particularly given the promising potential of neural network-based methods.

The neural network-based identification method we have tested excels in several aspects. It demonstrates exceptional specificity, effectively distinguishing autofluorescence and non-specific labeling from genuine neuronal staining. Notably, the neural network reduces the number of identified neurons that are inconsistent with manual analyses. While the initial time investment for training the neural network is substantial, the algorithm’s subsequent efficiency is evident as it can analyse hundreds of images within minutes without human intervention. Nevertheless, there are two notable limitations to consider. Firstly, it relies on the Olympus software, which entails a costly license. Secondly, the neural network does not classify objects as neurons but assigns a probability to each pixel in the image, indicating the likelihood of it being part of a neuron. To overcome these limitations, researchers can explore open-source software solutions such as StarDist [23], MIA [24], or BiaPy [25]. These open-source tools not only offer cost-effectiveness but are also well-suited for object identification. Furthermore, the flexibility to train the neural network under various conditions opens avenues for result improvement and application across diverse scenarios at minimal additional cost.

Although inconsistencies in neuron identifications emerged between observers and were particularly evident when compared to the semi-automatic macro initially employed in the Reigada et al. [16] study, the accuracy analyses conducted suggest that the estimates of the total number of neurons in each section remain reasonably consistent across various methods and observers. The factors that reconcile the divergent identifications when calculating the total number of neurons fall outside the scope of this paper. However, it is essential to underscore that these factors affirm the validity of the results presented in the original paper by Reigada et al. [16] despite the identification challenges associated with the method employed.

### 3.3. Analysis of Neuronal Death and Neuroprotection

In addition to exploring the potential for developing and testing image analysis tools, our objective was to leverage the stored images to investigate the distribution of neuronal death within the spinal cord in a mouse model of contusive SCI and identify the neurons protected by the treatment with the anti-apoptotic drug ucf-101. We focused on sections located 0.6 to 1.2 mm caudal to the injury epicenter, as this region initially showed the most pronounced neuroprotective effects of ucf-101 [16]. For neuron identification, we employed the neural network developed in this study. To ensure data comparability across different individuals and experimental conditions, we utilized thin plate spline techniques for image registration, aligning them with reference sections from the Mouse Spinal Cord Atlas by Watson et al. [18]. This approach facilitated not only the quantification of neurons but also their spatial localization and their allocation within Rexed laminae.

The findings from these analyses shed light on the distinctive pattern of neuronal death that unfolds the current model of SCI. Firstly, they confirm the existence of a gradient of neuronal death concerning the proximity to the injury epicenter, consistent with prior observations in Reigada et al. [16]. According to this pattern, the loss of neurons is massive at distances below 0.6 mm from the epicenter. This effect gradually wanes, ultimately resulting in more than 40% neuronal survival at a distance of 1.2 mm. We observed a similar pattern of neuronal loss within 400 microns caudal to the injury epicenter, with subsequent progressive survival as the distance from the epicenter increased in a recent investigation, examining neuronal survival following SCI in both wild-type (WT) and *Xiap* overexpressing C57BL/6 mice [26]. In a similar contusive model with higher severity (75 vs. 50 Kdynes), Sauerbeck et al. [27] observed a substantial number of neurons preserved in sections located 600 µm rostral to the injury epicenter in female mice. This contrasts with our current findings, which show that neurons are nearly absent at this distance. These disparities could suggest that caudal preservation is more constrained than rostral preservation, a possibility supported by our results in Reigada et al. [26].

Within the transverse section of the spinal cord, our results further suggest that cell death predominantly impacts neurons located in intermediate regions. Specifically, this phenomenon pertains to the nuclei and laminae situated adjacent to the ependyma, with a particular emphasis on laminae L5l, L5m, and L10, as well as nuclei D and LDCom. To what extent the observed pattern reveals the intrinsic vulnerability of neuronal populations or differences in the deleterious stimuli they confront is out of the scope of the present study. It is important to acknowledge that the consistency of this pattern within the present sample and across various studies remains open to question. Individual variability is substantial, and our current results allude to a general trend rather than a uniform observation.

Despite the extensive body of literature involving histological analyses of neuronal death in mouse models of SCI (a search on PubMed retrieves more than 480 articles on spinal cord injury AND neuronal death AND mouse), information on neuronal death distribution is almost lacking. In one of the few descriptions, Matsushita and colleagues [28] described the accumulation of TUNEL-positive neurons in the dorsal horns following ischemia/reperfusion SCI, in contrast to the accumulation of neuronal death in the intermediate region we have observed. While differences in the injury model may offer an explanation, the abundance of TUNEL-positive neurons in the most dorsal laminae may also reflect the higher density of neurons in this particular region.

In our 2015 study, we investigated the neuroprotective impact of ucf-101 in the context of SCI. Ucf-101 (5-[5-(2-nitrophenyl) furfuryliodine] 1,3-diphenyl-2-thiobarbituric acid) is a heterosynthetic cyclic compound known for its demonstrated neuroprotective effects in animal models of cerebral ischemia [29] and other neurodegenerative diseases [30,31]. According to our original study, the administration of ucf-101 resulted in a noteworthy increase in the survival of neurons located within the 0.6–1.2 mm caudal to the injury epicenter. Consistent with these findings, our current analysis corroborates ucf-101-induced neuroprotection in sections situated at 0.6 and 0.8 mm caudal from the injury epicenter. However, we have not detected significant neuroprotection at greater distances. These discrepancies arise from the advances in the methodology for neuron identification, particularly notable for sections located 1.0 mm caudal from the epicenter. In these sections, the threshold-based method employed in the original article identified multiple artifacts or background signals as neurons, which were excluded by the neural network-based method. Nonetheless, it is worth noting that the limited availability of samples with sufficient preservation for analysis at 1.2 mm may also contribute to the observed differences.

Our image analysis also allowed us to pinpoint specific Rexed laminae and nuclei that exhibit enhanced neuroprotection. In particular, we have observed increased survival of interneurons within laminae L4, L5l, L7, and the D and ICI nuclei. Interestingly, the increased neuroprotection within the intermediate region of the transverse section broadly agrees with the region of higher neuronal death following injury. Collectively, the spatial distribution of surviving neurons suggests that the protective effects of ucf-101 are primarily confined to neurons under strong deleterious pressures, particularly those closest to the injury epicenter. Considering the specific effects of ucf-101 on the intrinsic pathway of apoptosis, this observation could be interpreted as an indication that this pathway is active in the border of spared tissue surrounding the injury epicenter caudal to the injury. This observation aligns with previous clinical data and animal models of SCI (see [32] and references therein) and related conditions such as stroke [33].

The patterns of spatial distribution of neuronal death after SCI and the neuroprotective effects of ucf-101 described here can be considered, at best, as hypotheses to be tested. Variability among individuals and sections makes it difficult to draw more consistent patterns. A substantial part of the observed variability arises from biological or injury differences among individuals and constitutes key information to characterize the responses to injury and treatment, but variability is also a consequence of histological artifacts. For example, we observed a large difference in the number of neurons identified in three sections from one control individual (216, 260, and 503 neurons), which was associated with a high staining intensity in one of the sections. To address variability in neuronal counts within conditions, we have employed the median as a centrality measurement through most analyses. However, further analysis and quality controls may be required to deal with histological artifacts. On the contrary, dealing with biological variability will benefit from increasing the sample and incorporating additional information on the injury. Our intention for future analyses is to incorporate samples from additional studies carried out by the group [26,34] as well as additional sources of information (e.g., tissue sparing) for characterizing the variability of neuronal death through individuals and studies.

Several steps of the present image analysis can be improved. The first one concerns image registration, the process of overlaying two or more images to achieve maximum correspondence, which we carried out using BigWarp, a tool for landmark-based deformable image alignment [35]. A number of image registration methods have been proposed, including several specifically developed for histological images (for example, cWR [36]). However, in our hands, neither this nor more general tools, such as pyStackReg [37] or Elastix [38], yielded better results than BigWarp. Morphological deformations, together with staining differences, make automatic registration of histological sections highly challenging. Indeed, a recent challenge demonstrated that image registration of histological sections requires the development of specific tools, which complicates or even makes it impossible for regular users to apply those [39]. In all, results from image registration using BigWarp can be considered optimal, although several aspects should be improved. On the one hand, landmarks employed for thin plate spline deformations were not standardized, but both their number and position were selected subjectively to obtain the best alignment between the section under study and the reference Mouse Spìnal Cord Atlas [18]. On average, we employed 35 landmarks; however, the number varied based on the preservation of the spine; well-preserved gray substances required few landmarks (down to a minimum of 11), while highly deformed ones required up to 53 landmarks to obtain an acceptable alignment. Moreover, difficulties in the identification of homologous positions of landmarks in some damaged sections also made deformations (and, therefore, the subsequent quantifications) questionable. Future studies should focus on defining a set of identifiable landmarks warranting homologous positions in the reference and moving image. On the other hand, we employed the Mouse Spinal Cord Atlas by Watson et al. [18] as the moving image that was aligned to the image under analysis. While this approach ensures that the section and its component neurons are not deformed, it forces manual counting of the neurons in each nucleus and lamina. This step can be automated if we employ the spinal cord section under analysis as the moving image (instead of the reference map); in this way, the regions of interest (ROIs, i.e., laminae and nuclei) defined in the reference image can be automatically quantified. Recently, Fiederling et al. [15] have described procedures for neuronal quantification based on registering 3D reference volumes derived from the Allen’s Mouse Spinal Cord Atlas [40] instead of the segment 2D annotations of this Atlas. In future studies, we will test whether this approach solves the registration problems we have identified.

A final limitation of our methods concerns the quantification of neurons in each Rexed laminae or nuclei. Even though this regionalization has well-established biological roots, each region having specific features in terms of neuronal function, chemistry, and morphology, this way of analysing the spatial distribution of the surviving neurons has limitations. On the one hand, the boundaries of laminae and nuclei are section specific and, therefore, the assignment of neurons, in cases such as the intermediolateral nucleus, can be questionable when using a reference Atlas that is derived from a different section of a different individual. On the other hand, the use of predefined regions restricts the spatial changes in neuronal death to those detectable at this regional scale, i.e., patterns of neuronal survival/cell death unrelated to the proposed regionalization are lost or at least diluted. In this respect, we consider that direct analysis of the spatial location of each surviving neuron through spatial statistics approaches will be more suitable for detecting patterns or correlations of neuronal death within the spinal cord as well as for reducing subjectivity. In this respect, the procedures employed in the Blue Brain project [41], by defining the voxels in a reference volume as the basic units for quantifying neuronal abundance, can be a valid alternative.

### 3.4. Perspectives

In the last decades, significant efforts have been made to repair SCI. Our understanding of SCI pathophysiology has expanded, and multiple therapeutic treatments have been developed, showing varying degrees of success in animal models. Despite these advancements, we have yet to discover a single therapy that demonstrates both efficacy and safety in clinical trials, qualifying it for clinical practice. This study not only contributes to the understanding of SCI but also aligns with an exciting area where the integration of advanced techniques, such as 3D mapping of neurons based on image analysis and the revolutionary Single Cell RNA sequencing, is redefining the boundaries of neuroscience research. Leading initiatives, such as the Blue Brain Project, have been at the forefront of this movement. The combined analysis of spinal cord images with single-cell RNA sequencing data opens new avenues for addressing fundamental questions in neurobiology and lays the groundwork for future clinical applications. However, there is a noticeable absence of information concerning the spatial distribution of the different forms of neuronal death over time. Available spatial information dates back to the 1990s and the early 2000s and is restricted to necrotic and apoptotic cell deaths, particularly in rat models of SCI (see, for example, [42]). On the contrary, thousands of histological preparations from mouse models of SCI are employed in all but a few articles to illustrate differences among conditions or to compare specific aspects of these conditions. In light of ethical imperatives to minimize the use of laboratory animals, we have carried out a proof-of-concept to explore the potential of leveraging recent technological advancements to maximize the wealth of information derived from the histological data gathered through the years. Our results, available at the neuroCLUEDO repository, confirm that previous histologies preserve information key to decoding and understanding the processes responsible for the spatial distribution of neuronal death and the neuroprotective effects of therapies under development.

## 4. Materials and Methods

### 4.1. Images

The dataset comprises 168 confocal images from full cross-sections of female C57BL/6J mice spinal cords up to 1.5 mm rostral and caudal to the injury epicenter. Images correspond to sections of spinal cords from two control (laminectomized without injury) and eleven injured (50 Kdynes moderate contusion at spinal segment T11 using an Infinity Horizon Impactor from Precision Systems and Instrumentation (Lexington, KY, USA)) individuals (five treated with ucf-101 and six with vehicle). These images were originally employed to analyse the neuroprotective effects of the drug ucf-101 [16]. Injured and control animals were euthanized 21 days after surgery. After processing, their spinal cords were cut into 20 µm thick transversal sections. For neuronal quantification, sections were stained with an antibody against the neuronal marker NeuN (RRID: AB_11210778, Millipore cat#MAB377, Burlington, MA, USA) and the nuclei marker DAPI (4′,6-diamidino-2-phenylindole, Sigma Aldrich, St Louis, MO, USA) and imaged with a Leica (Wetzlar, Germany) TCS SP5 fast-scan confocal microscope equipped with a PL APO 20×/0.70 CS 20X objective.

Additional information can be accessed in the original publication [16] and at OSF (https://osf.io/qe9ys/; accessed on 7 April 2025). The repository includes confocal images in .lif format plus metadata for each individual. Each .lif file contains all imaged sections from an individual. Metadata files comprise Minimum Information about Spinal Cord Injury Experiments (MIASCI) data, as well as information on each animal, section, and image contained in the .lif files.

For all analyses in this study, we have transformed the confocal images from each individual stored in each .lif file into independent .tif files. The transformation was carried out using Fiji (ImageJ ver. 1.53c [43]) and involved maximum intensity projection of the focal planes of the confocal image and its conversion into RGB format.

### 4.2. Neuron Identifications

Three identification methods were compared: one based on manual identifications, another based on semi-automatic thresholding, and the last based on a deep-learning neural network. The experimental design is summarized in Figure 7. Out of the 168 images stored at OSF, 87 were employed for training and 20 images for comparison of neuron identification methods. Both the total number of identified neurons in the section and the position of each identified neuron were employed in these comparisons. Up to five manual quantifications per image were employed to obtain a consensus estimate of the number of spinal neurons in each section of the comparison set.

The comparison set of images was analysed using the following: (i) a manual analysis carried out by six observers using the Cell Counter plugin from Fiji; (ii) a threshold-based identification with a custom Fiji macro [16]; (iii) a neural network-based identification employing the deep-learning-based image analysis approach TruAI integrated CellSens Dimension software (Olympus, L’Hospitalet de Llobregat, Spain).

The neurons identified by each method were compared by overlapping their positions or segmentations obtained with each method for each image. Repeatability, reproducibility, and accuracy of quantifications of the total number of neurons per section were estimated. For details, see Appendix B.

### 4.3. Reanalysis of ucf-101 Effects on Neuronal Survival

In total, 41 sections were included to reanalyse the effect of ucf-101 on neuronal survival. Neurons were identified using the neural network-based identification indicated above. With the aim of obtaining an estimation of the neurons present in each Rexed lamina and nucleus of the analysed sections, images were registered to a reference Atlas of the thoracic segment 11 [18]. For details, see Appendix B.

## Figures and Tables

**Figure 1 ijms-26-03749-f001:**
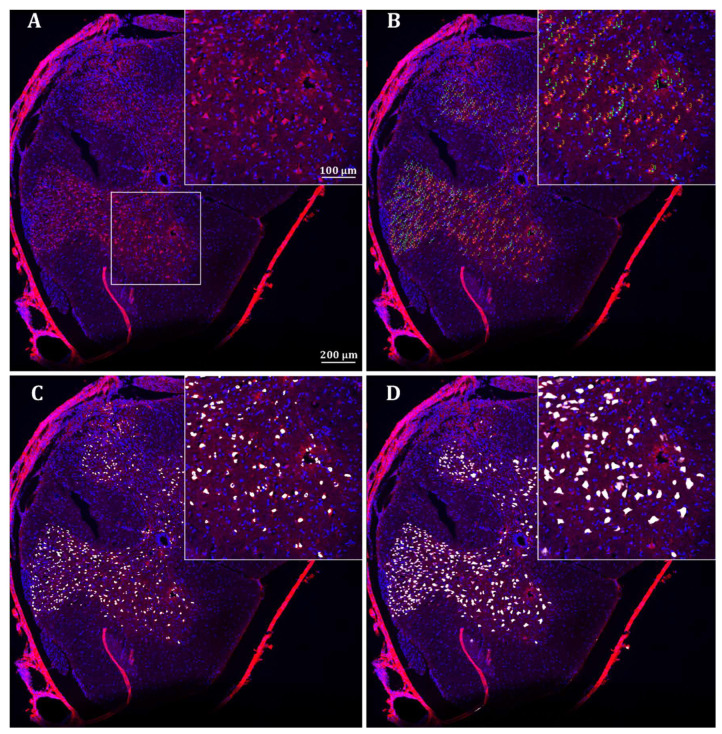
Neuron identifications in the spinal cord. The figure shows the identifications obtained with the three methods in the full section and a detail (insert) of image 197.tif. (**A**). Image under analysis. (**B**). Manual identifications. Numbers represent the number of times each neuron was identified after five analyses. (**C**). Obtained segmentations with the threshold-based method. (**D**). Probabilities of each pixel to be part of a neuron (in a 0 to 255 intensity scale) according to the neural network. Red color corresponds to NeuN staining, blue to DAPI, and white to the outputs of the different identification methods. Scale bars in all sub-figures are the same as in (**A**).

**Figure 2 ijms-26-03749-f002:**
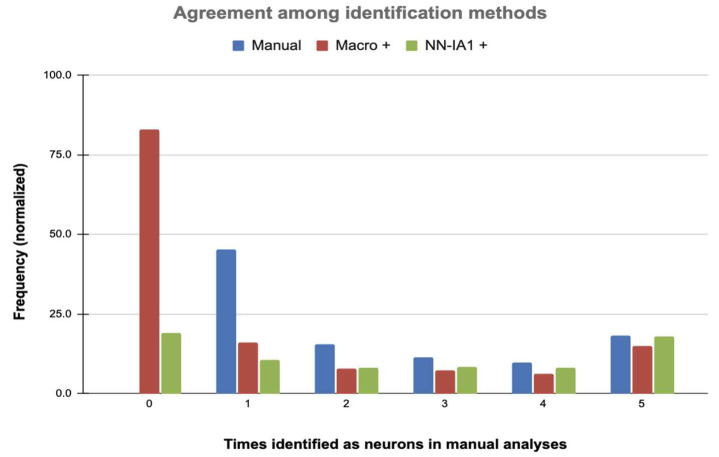
Agreement between manual, threshold, and neural network-based identifications. The graph represents the overlapping of the objects identified as neurons by the three methods in five spinal cord sections. Classes 0 to 5 in the x-axis indicate the number of times an object was identified as a neuron after five manual analyses. Class 0 corresponds to objects identified as neurons by the threshold and/or the neural network methods that were not identified as such in any manual analyses. Bars represent the mean frequency of each class after normalization to 100 neurons in manual analyses. Full data are available in Appendix A.

**Figure 3 ijms-26-03749-f003:**
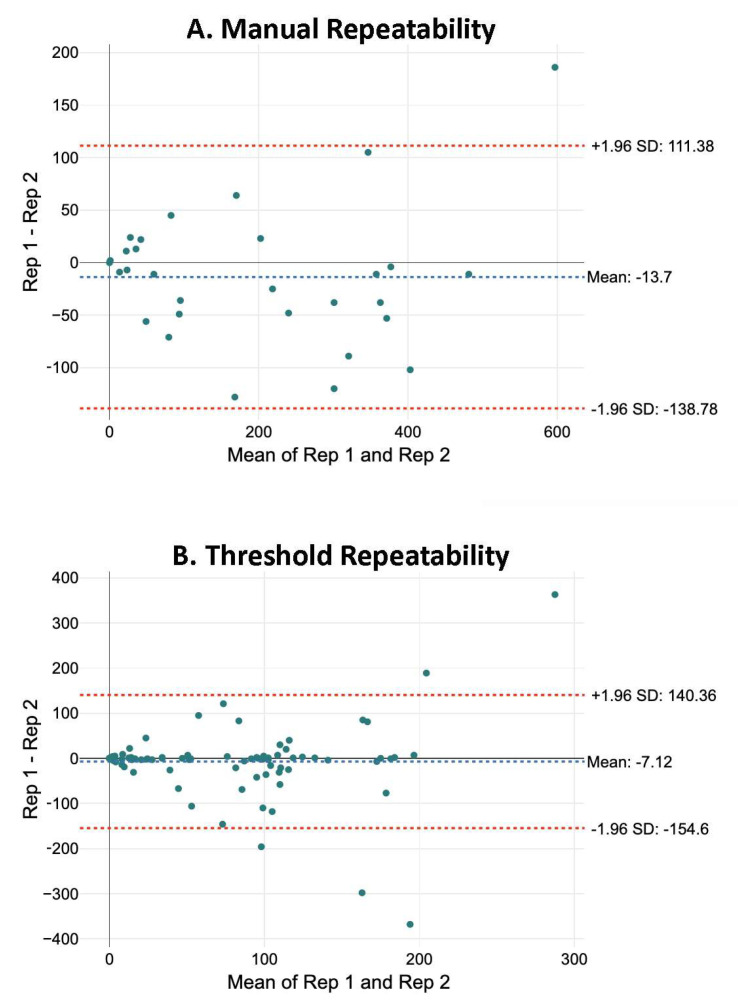
Repeatability of identifications. Bland–Altman plot depicting the repeatability of replicate identifications by observers/analysts. The graph compares the differences between repeated counts of the total number of neurons in a given section versus their mean. Counts were repeated blindly, with at least one week between counts. Red horizontal lines refer to 95% confidence intervals. (**A**): repeatability of manual counts (*n* = 30 repeated counts). (**B**): repeatability of threshold-based counts (*n* = 100 repeated counts).

**Figure 4 ijms-26-03749-f004:**
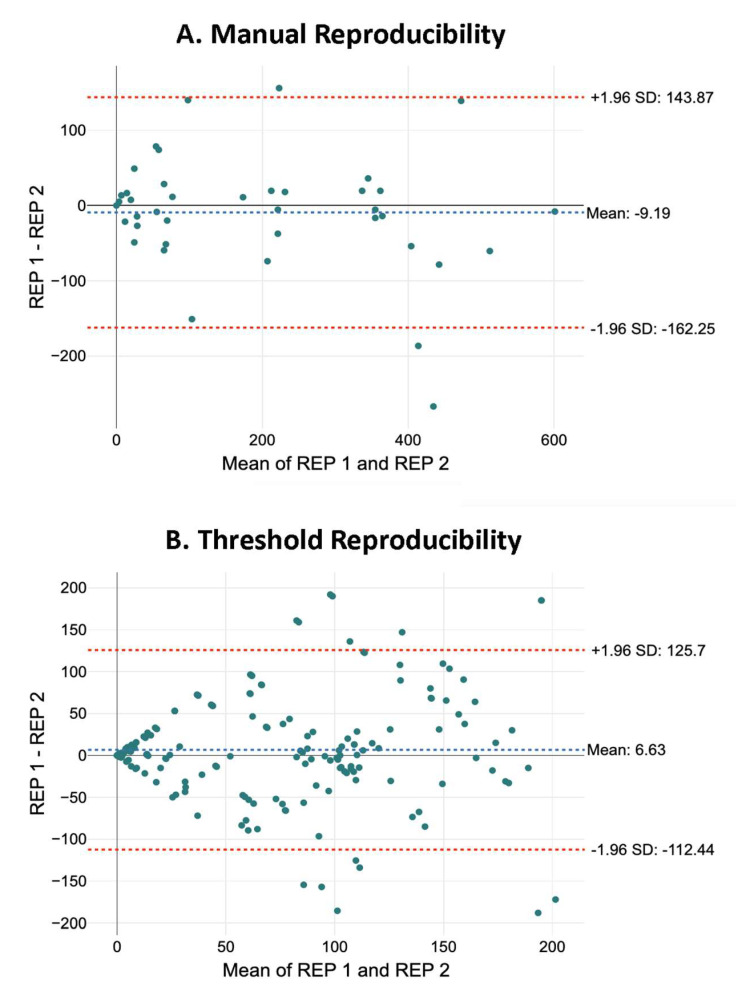
Reproducibility of manual identifications. Graphical representation of reproducibility according to the Bland–Altman plot. The graph compares the differences in the number of neurons counted by two analysts with the mean of both values. Red horizontal lines correspond to the ±95% confidence intervals. (**A**): repeatability of manual counts (*n* = 40 repeated counts). (**B**): repeatability of threshold-based counts (*n* = 200 repeated counts).

**Figure 5 ijms-26-03749-f005:**
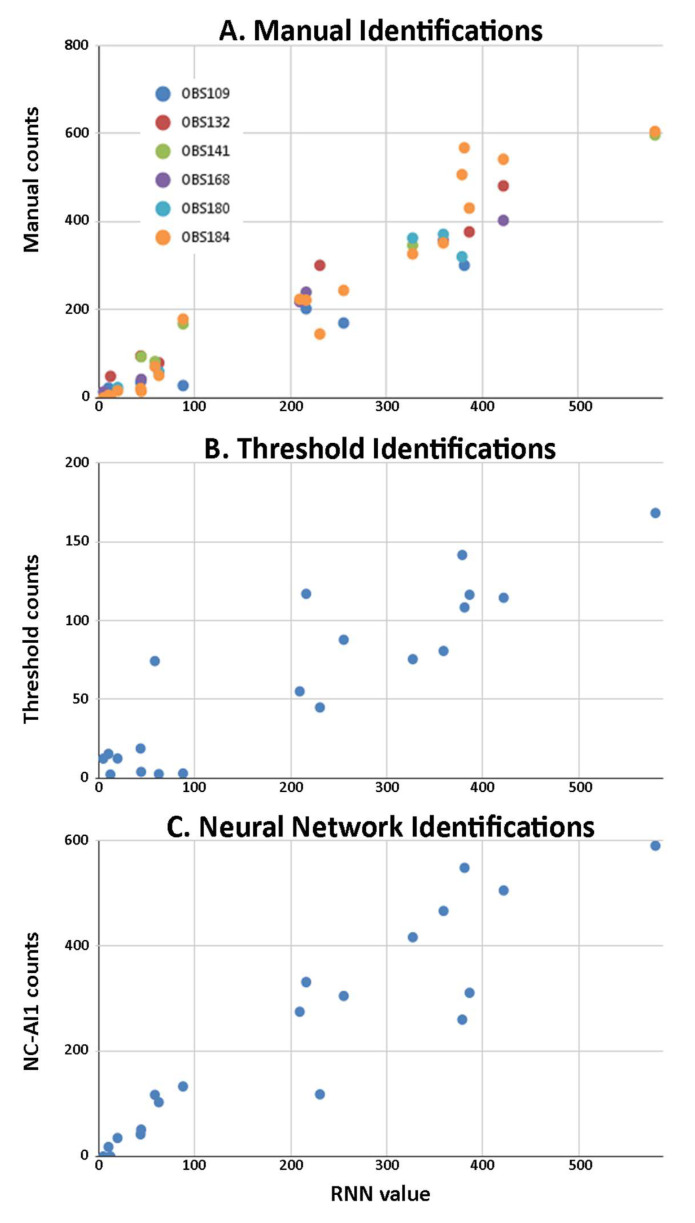
Accuracy of neuronal counts from manual, threshold-based, and neural network-based identifications. Comparison of the Reference Number of Neurons (RNN) with the estimated values in manual observations (**A**), threshold-based identifications (**B**), and neural network (NC-AI1)-based identifications (**C**). The symbols and colors in (**A**) correspond to each of the observers.

**Figure 6 ijms-26-03749-f006:**
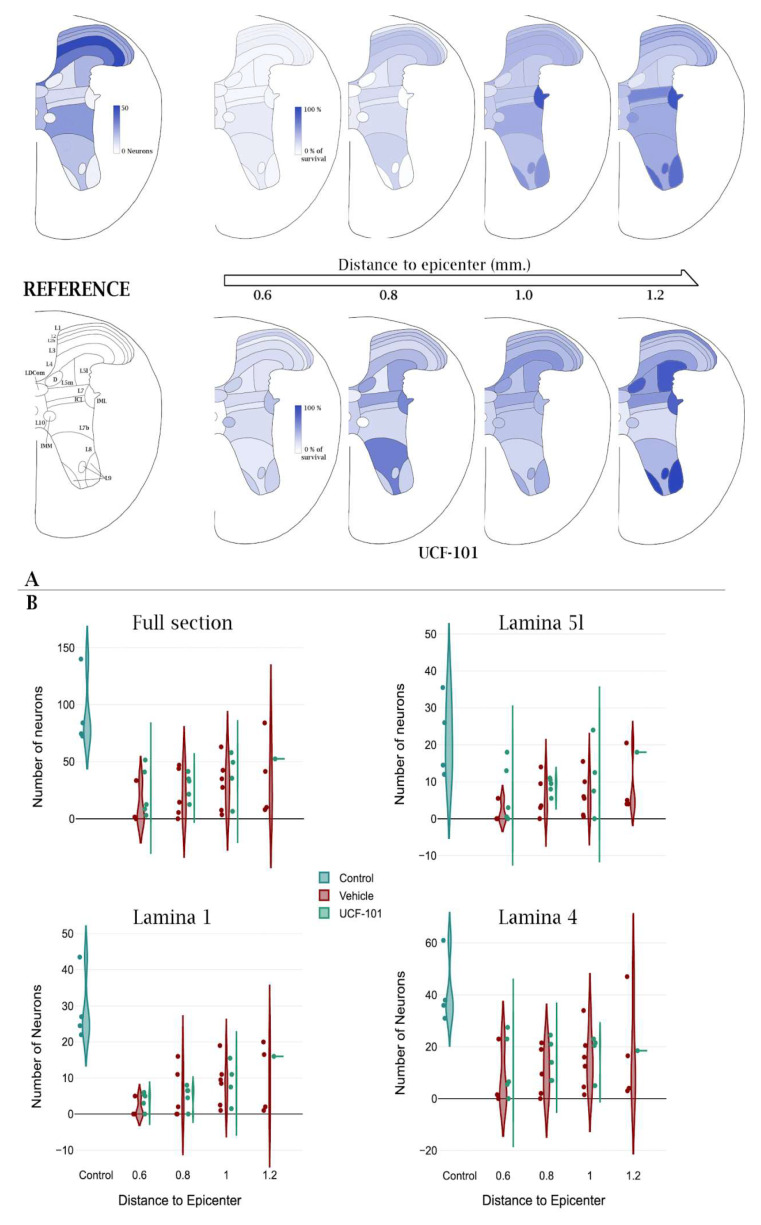
Changes in the number of neurons after SCI and ucf-101 treatment. (**A**). The spinal cord drawings illustrate the number of neurons in each Rexed lamina of the T11 segment of the naïve spinal cord (Control) and the percentage of surviving neurons (relative to the naïve spinal cord) in each lamina at 0.6, 0.8, 1.0, and 1.2 mm caudal to the lesion epicenter after SCI plus vehicle (above) or ucf-101 treatment (below). The graph also illustrates Rexed laminae and nucleus according to the Mouse Spinal Cord Atlas by Watson et al. [18]. Data corresponding to the median values of the different individuals and sections analyses are available in Appendix A. The spine image with the annotation of Rexed laminae was modified from the Mouse Spinal Cord Atlas by Watson et al. [18]. (**B**). Violin plots depicting the number of neurons in the whole section and in selected laminae of naïve, injured, and ucf-101-treated spinal cords.

**Figure 7 ijms-26-03749-f007:**
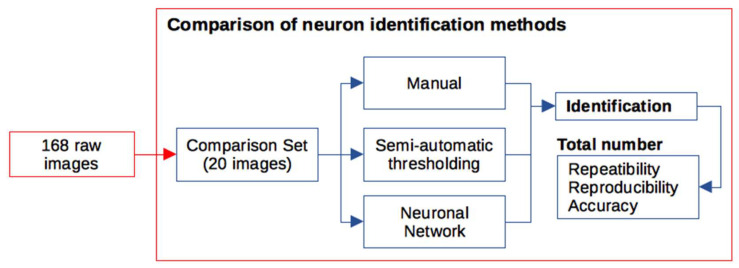
Summary of the study design. In total, 168 images of full spinal cord sections from 11 control, injured, and treated mice were deposited at OSF. Of these images, 20 were selected as the comparison set. Data on the identified neurons and their total number per section were employed to compare manual, threshold-based, and neural network-based methods of neuronal identification.

**Table 1 ijms-26-03749-t001:** Contents related to this study of the OSF repository (https://osf.io/n32z9/; accessed on 7 April 2025). The table lists and describes all files related to this study present in each of the three components of neuroCLUEDO. Additional contents related to other analyses can be present in each component within specific subcomponents. Each component and subcomponent has its own wiki page with additional details and information.

Component	Files		Description
Image Repository: ucf-101 files (https://osf.io/qe9ys/; accessed on 7 April 2025)	.lif files (13)		Each .lif file comprises all confocal images (stacks) of the spinal cord sections from one individual (in .lif format).
	2015 Reigada.pdf		The original article where images were published (.pdf format).
	metadata images.xlsx		Spreadsheet with data on each section (individual, treatment, number of Z, rostrocaudal position, etc.; in .xlsx format).
	MIASCI data.xlsx		Minimum information about spinal cord injury experiments of the study (in .xlsx format).
Test Benches/ Neuron Detection/ TestBench1. ucf-101 images (https://osf.io/7agyq/; accessed on 7 April 2025)	tif images folder		A folder comprising two zip files that contain the images (in .tif format) of the comparison (20 images) and training sets (87 images).
	Manual Identifications Folder		
		Manual ID_Spreadsheet for data collection.xlsx	Excel spreadsheet employed by the observers for data collection (in .xlsx format). Includes instructions.
		Manual ID_identifications xml.zip	A file containing folders for each image with the image, identifications of each observer, and the overlap of all of them. Identification data are presented in .xml format exported from the count tool of Fiji ver. 2.16.0/1.54p.
	Threshold Macro Nukleator Folder		
		Threshold ID_macro.txt	ImageJ/Fiji macro for neuron thresholding developed and employed in the Reigada et al. [16] study (in .txt format).
		Threshold ID_idetifications.zip	zip file containing 20 files with the neuronal identifications derived obtained with the threshold macro (in .tif format).
	Neural Network NC-AI1 Folder		
		Neural Network_NC_AI1.nn	Neural network developed in this study. In .nn format of CellSens Dimension (Olympus) ver. 3.1 software. To be used with CellSens Dimension software.
		NC-AI1 Training.zip	A zip-compressed folder comprising the different types of files employed to train the neural network (in .ets format).
		NC-AI1 Identifications.zip	A zip-compressed folder containing the files of the comparison set with the neural network identifications (in .tif format).
		NC-AI1 ImageJ macro.ijm	Macro to identify as neurons those particles above 25 µm^2^ composed of pixels above 50% probability according to the neural network (in ImageJ’s .ijm format).
	Ibañez 2021_Degree Project_in spanish.pdf		Final degree project of Nadia Ibañez-Barranco describing the original results of this test bench (in .pdf format, in Spanish).
	Analysis		
		Overlapping.zip	A file containing folders for five images with the image identifications of all observers and the overlap with threshold and neural network identifications. Identification data are presented in xml format exported from the count tool of ImageJ or Fiji.
		Data and Analyses.xlsx	Excel file with all data and analyses from the comparisons of the manual, Nukleator, and NC-AI1 methods
Neuronal death reconstructions: Which neurons protect ucf-101? (https://osf.io/fhxqs/; accessed on 7 April 2025)	Processed images.zip		A folder containing the images with layers detailing neuronal identifications and annotations of Rexed laminae from Watson et al. [18]’s Atlas (in .tif format). This includes files (in .xml format) containing the landmarks employed for image registration.
	Software_bigdataviewer_fiji-6.2.1.jar		Fiji vs 2.16.0/1.54p plugins for image registration using BigWarp, a tool for landmark-based deformable image alignment (in .jar format).
	Software_bigwarp_fiji-7.0.2.jar		
	data &analyses.xlsx		Metadata, raw and processed data on the numbers of neurons *per* Rexed laminae and results of their analysis (in .xlsx format).
	R Analysis Neuronal Death.txt		Statistical analysis of the data in R ver 4.4.0 (in .txt format)

## Data Availability

Raw and processed images, software, macros, scripts, data matrices, and results are available in the Open Science Framework (OSF) repository neuroCLUEDO (https://osf.io/n32z9/; accessed on 7 April 2025).

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
