# Peer review of "Reanalysis of Published Histological Data Can Help to Characterize Neuronal Death After Spinal Cord Injury"

_ijms, 2025, doi:10.3390/ijms26083749_

Round 1
Reviewer 1 Report
Comments and Suggestions for Authors
The manuscript entitled “Reanalysis of published histological data can help to characterize neuronal death after Spinal Cord Injury” presents a compelling and well-executed approach to re-evaluating previously published histological data using a combination of manual, semi-automated, and deep-learning methods for neuronal counting.
The work is scientifically sound, well-structured, and written in clear English. The development of the neuroCLUEDO repository, and its application for benchmarking neuronal death estimation methods, adds a significant contribution to the field of spinal cord injury (SCI) research and to open science practices more broadly.
The authors have thoroughly described their methodology, provided appropriate statistical validation, and openly shared data and analysis tools via OSF. Their comparison of counting strategies—including the use of a convolutional neural network (CNN)—is both timely and relevant, especially for the reanalysis of historical datasets where manual quantification is not always feasible.The discussion of biological variability across spinal cord sections and the limitations of histological artifacts is thoughtful and honest. The potential application of this approach to reevaluate previous findings, such as the effect of ucf-101 treatment, is very interesting and demonstrates the power of reproducible and automated methods in experimental neurology.
Author Response
Dear Reviewer,
We just want to thank a lot your supportive comments.
All the best
Reviewer 2 Report
Comments and Suggestions for Authors
The manuscript report on an indepth technical analysis of the outcome for the count of immunolabeled neurons in a model of SCI spinal cord. Authors compare the manual evaluation with two semiautomathed methods, namely a threshold analysis carried out with Image J and a network analysis carried out with an Olympus software trained to recognize neuronal cells on the basis of their size. To compare the three methods thay use the ucf-101, a marker of neuronal death, that has been previously thouroughly characterized by the same group.
The most valuable contribution of this study for the study of neuronal death after SCI is the creation of an open-access repository hosted on the OSF platform. The authors point out very carefully pros and cons/limitations of the study. One of the latter ebing represented by the fact the the repository is limited to local data, however promising to be expanded to collect measurment data from the international groups working on this topic.
The manuscript wants to share with the scientific community the concept of uniformity of data interpretation among the different research group and the fact that the automathed method outperforms the other two methods in precision and reproducibility.
So, considering that Introduction, Material and Methods, Results, Figures and Discussion are very clearly delineated and the References are adequate, the authors deserve respect and congratulations for the excellent work.
Author Response
Dear reviewer,
We are really pleased you enjoyed our work. Many thanks for your supportive comments.
All the best